# Mechanisms Underlying Cell Therapy in Liver Fibrosis: An Overview

**DOI:** 10.3390/cells8111339

**Published:** 2019-10-29

**Authors:** Daphne Pinheiro, Isabelle Dias, Karina Ribeiro Silva, Ana Carolina Stumbo, Alessandra Thole, Erika Cortez, Lais de Carvalho, Ralf Weiskirchen, Simone Carvalho

**Affiliations:** 1Laboratory of Stem Cell Research, Histology and Embryology Department, Biology Institute, State University of Rio de Janeiro, Rio de Janeiro 20550-170, Brazil; daph.note@gmail.com (D.P.); zabelle.dias@gmail.com (I.D.); ribeiro.ks@gmail.com (K.R.S.); stumbo.ac@gmail.com (A.C.S.); alethole@uol.com.br (A.T.); cortez.erika@gmail.com (E.C.); ldc29@globo.com (L.d.C.); 2Institute of Molecular Pathobiochemistry, Experimental Gene Therapy and Clinical Chemistry, RWTH University Hospital Aachen, D-52074 Aachen, Germany; rweiskirchen@ukaachen.de

**Keywords:** liver fibrosis, cell therapy, stem cells, progenitor cells, hepatocytes, secretome, in vitro

## Abstract

Fibrosis is a common feature in most pathogenetic processes in the liver, and usually results from a chronic insult that depletes the regenerative capacity of hepatocytes and activates multiple inflammatory pathways, recruiting resident and circulating immune cells, endothelial cells, non-parenchymal hepatic stellate cells, and fibroblasts, which become activated and lead to excessive extracellular matrix accumulation. The ongoing development of liver fibrosis results in a clinically silent and progressive loss of hepatocyte function, demanding the constant need for liver transplantation in clinical practice, and motivating the search for other treatments as the chances of obtaining compatible viable livers become scarcer. Although initially cell therapy has emerged as a plausible alternative to organ transplantation, many factors still challenge the establishment of this technique as a main or even additional therapeutic tool. Herein, the authors discuss the most recent advances and point out the corners and some controversies over several protocols and models that have shown promising results as potential candidates for cell therapy for liver fibrosis, presenting the respective mechanisms proposed for liver regeneration in each case.

## 1. Introduction

The marvel of tissue regeneration has fascinated civilizations as ancient as the Greeks, who perpetuated the belief in liver regeneration based upon the Prometheus’ myth. Therefore, the knowledge and implications of the liver’s capacity to regenerate itself has been a longstanding theme in scientific research. Despite this remarkable feature, liver failure is still a major health problem worldwide. Currently, molecular mechanisms and cell pathways involved in the exhaustion of liver regenerative potential are well known, and factors such as the chronicity of the insulting agent are key elements in this cycle of tissue inflammation and burnout. As an extremely complex organ, it is assumable that different cell processes interact in a specific time and sites in the liver to trigger the alterations that lead to tissue fibrosis, metabolic imbalance, loss of hepatic function, and increased tumorigenic potential [1,2].

As the current issue addresses these aspects, this chapter will focus on the current methodologies being developed to overcome the common end of all these adverse changes: liver failure. This brings us to the intrinsic problem of the shortage of available and viable organs for donation, late diagnosis, and the systemic immune consequences of unexpected incompatibility after transplantation, all of these resulting in high morbidity and mortality levels. In the Western world today, there are about 70,000 patients with end-stage chronic liver disease, and in this group the mortality rate is up to 80%. Therefore, in the era of regenerative medicine, a myriad of approaches have rapidly emerged in an attempt to minimize the poor outcomes of chronic liver disease and to extend the lifetime for patients awaiting liver transplantation, with different mechanisms, technologies, and results obtained, according to different protocols and disease aspects [3]. The next sections will explore the most promising therapies proposed to the date.

Main liver parenchymal cells, the hepatocytes, not only play key roles in digestive, endocrine, circulatory and secretory body functions, but also display a remarkable regenerative potential. However, continued injury caused by toxins or inflammatory factors leads these cells to a chronic oxidative stress that can trigger cell cycle arrest and exhaustion of the metabolic machinery. Injured hepatocytes secrete cytokines, such as tumor necrosis factor alpha (TNF-α), that attract other immune and inflammatory cells. Once these cells respond, secreting great amounts of transforming growth factor beta (TGF-β) and platelet derived growth factor (PDGF), liver cells that synthesize the normal extracellular matrix (ECM), such as perisinusoidal hepatic stellate cells (HSCs) and periportal fibroblasts, start their differentiation into myofibroblasts [4]. These are fibrogenic cells, which produce and deposit a pathological collagen-rich ECM in perisinusoidal and periportal areas. These activated cells are a heterogeneous population with different subsets that share the common expression of alpha-smooth muscle actin (α-SMA) and the marker S100A6 [5].

The main consequence of continued liver damage is fibrosis development, which progressively decreases blood flow to hepatocyte plates, leading to increased hypoxia and cell death. Liver disease progression can take a variable amount of time and spread, depending on the nature and frequency of the insult, but is clinically silent. Most patients who experience hepatic symptoms, receive as their first diagnosis the report of a widespread and advanced fibrosis state, known as cirrhosis. Although hepatic fibrosis is reversible, cirrhosis has a poorer prognosis and often leads to liver failure and the need for orthotopic liver transplantation at some point in the patient’s treatment. Therefore, end-stage liver disease still defies health care systems throughout the world [6].

Several factors may cause hepatocyte injury, among them the Western hypercaloric diet associated with non-alcoholic fatty liver disease, infections with hepatitis C virus, misapplication of pharmacological substances, drug abuse, liver malignancies affecting hepatocytes (hepatomas) or the biliary intrahepatic/extrahepatic tract (cholangiocarcinomas), and congenital or acquired cholangiopathies affecting normal bile flux [7,8].

Liver disease modeling includes animal, ex vivo, and in vitro models, which are to be chosen according to each kind of hepatic disease and considering the extremely complex nature of the organ. Most common animal models of toxic injury use a series of administrations of hepatotoxic agents, such as CCl4 or acetaminophen and its derivatives in rodents. The most common surgical model is bile duct ligation (BDL), with or without bile duct resection [9]. This leads to bile accumulation in the liver and therefore is suitable for the study of cholestatic diseases, such as bile duct atresia, bile duct adenocarcinoma, or gallstones. There is some evidence that some BDL procedures can be spontaneously reversed in rats, and this may be followed by liver regeneration. A safer and less invasive alternative to the BDL model is the 3,5 diethoxicarbonyl-1,4 dihydrocollidine (DDC)-diet, which also induces reversible cholestasis [10].

The increasing demand for the study of metabolic chronic diseases, especially in the context of the hypercaloric diet consumed in Western countries and the development of obesity, diabetes, cardiovascular disease, and systemic metabolic alterations, has also impacted the hepatology field. Because the liver has a central role in metabolism, dietary habits have a direct impact on this organ, and metabolic syndrome is often associated with abnormal lipid accumulation in hepatocytes, in a condition known as nonalcoholic fatty liver disease (NAFLD). NAFLD has become a major concern because of its dramatic increase in the past decades, being the most common liver disease in many countries. There are different and progressive levels of NAFLD, when many factors or “multiple hits” of metabolic and inflammatory nature lead to more severe alterations. These may vary from simple steatosis with few clinical relevance, to widespread macro-steatosis accompanied by inflammatory lesions, known as nonalcoholic steatohepatitis (NASH), that can lead to fibrosis, cirrhosis, and hepatomas [11,12]. Experimental protocols for NAFLD studies in animals commonly include a high-fat diet, high-fructose diet, or methionine and choline-deficient diets. Genetic models are based on leptin (*ob/ob* or *db/db*), peroxisome proliferator-activated receptors (PPARs), or CD36 deficiencies, and often need a second injury or “hit” (usually dietary models). There are also toxic and pharmacological interventions with porphyrinogenic agents, lipopolysaccharide, or streptozotocin. Recent studies usually combine two or more models to mimic human NASH [13].

Animal models have limitations because they do not exactly reproduce human liver diseases, and rodents have different regenerative potentials. Common cell models include manipulation of hepatocyte cell lines, usually derived from human hepatomas (HepG2, Huh-7). Most recently, with the development of cell reprogramming technology [14], hepatocyte progenitors can be obtained, expanded, and differentiated in vitro, as well as other liver parenchymal cells [15].

The main goal of cell therapy in liver fibrosis is to stop its progression and engage the necessary factors for fibrosis resolution in most organs: myofibroblast apoptosis and/or deactivation/senescence, ECM remodeling, and parenchymal stimulation in a paracrine way [16]. This is the point where cell therapy can play an important role, along with other promising pharmacological and surgical advances in an interchangeable combination that can achieve the turning point for fibrosis regression [17].

## 2. Cell Types Used for the Inhibition of Liver Fibrosis

Cell therapy is an expanding research field and refers to a treatment wherein any cell type and/or cell product—which can also be modified by biotechnology techniques—is targeted for human, animal, or in vitro models of disease. The main goals of cell therapy are to reach continued and stable tissue regeneration, with minor or negligible adverse effects [18]. Future therapeutic schemes will probably focus on a combination of cell therapy and conventional (pharmacological and/or surgical) treatment plans for the best clinical practice.

The most usual drawbacks of cell therapy are frequently a consequence of the developing nature of the field, and recent works focus on reaching a personalized scheme for each patient and disease. This can result in very complex combinations regarding cell types and products used, origin (endogenous or exogenous), additional manipulation in the laboratory, dosage, single, or multiple doses, route of administration, time of intervention, and comorbidities. All these factors can result in higher costs and standards of care to enable the clinical use of cell therapy [19]. Still, once positive outcomes are reached by a refined technique in preclinical phases, cell therapy is likely to be cost-effective and beneficial.

### 2.1. Bone Marrow Mononuclear Fraction

Adult bone marrow is defined as a specialized niche intimately related to blood, bone, and vascular tissues. Two main subtypes of stem cells reside in the bone marrow stroma: (a) hematopoietic stem cells, which form all blood cell lines and tissue phagocytes, and (b) mesenchymal stem cells, responsible for hematopoietic stem cell survival, renewal, and maintenance [20]. Bone marrow transplantation is among the earliest cell therapy techniques developed and has been extensively used to treat leukemia and immunological deficiency disorders for several decades, being the choice treatment for acute myeloid leukemia. The transplantation is effective for colonizing and restoring bone marrow function in patients after radiotherapy [21]. In the last years, studies have shown that bone marrow mononuclear cell (BMMN) transplantation has therapeutic effects on liver function and ameliorates fibrosis (Table 1).

Several aspects of liver function restoration in cholestatic fibrosis after BMMN therapy have been studied by our group over recent years. Using this model, we observed a reduction in collagen types I and IV, laminin, and in the number of bile ducts [22]. BMMN transplantation promotes activation and increase in matrix metalloproteinase-9 (MMP-9) and -13 (MMP-13) levels, while decreasing tissue inhibitor of metalloproteinase (TIMPs) expression and promoting fibrogenic cells apoptosis, all these factors contributing to ECM degradation [23,24]. BMMN therapy also improves mitochondrial bioenergetics through stimulating liver oxidative capacity, reducing oxidative stress, and also regulating mitochondrial coupling (UCP2 levels) and biogenesis (PGC1-α levels) in fibrotic livers, indicating a metabolic recovery of the organ [25].

In addition, BMMN transplantation in cholestatic mice decreased Kupffer cells (CD68^+^) and increased neutrophils (Ly6G^+^) numbers in fibrotic livers. After transplantation, BMMNs successfully engrafted in the fibrotic liver, directly contributing to the total populations of endothelial cells (CD144^+^), extrahepatic macrophages (CD11b^+^), and neutrophils (Ly6G^+^). These events were positively correlated to the augmented expression of anti-fibrotic cytokines (IL-10, IL-13 and hepatocyte growth factor, HGF) and the reduction in pro-inflammatory cytokines (IL-17A and IL-6), indicating a shift in the cytokine expression pattern and in macrophage activity/phenotype, from M1 proinflammatory cells to a M2 anti-inflammatory and pro-fibrolytic phenotype [26]. Similar results were found after CD11b^+^CD14^+^ bone marrow monocyte therapy, which decreased pro-inflammatory cytokines IL-6 and IL-1β, TGF-β1, and TIMP-1, while IL-10 and MMP-9 were increased [27]. The hypothesis for liver regeneration in this scenario is explained in Figure 1. In another study, Suh and colleagues transplanted CD11b^+^Gr1^+^F4/80^+^ monocytes into fibrotic mice and found that this increased CD4^+^CD25^+^Foxp3^+^ Treg cells and IL-10 production, decreasing IL-6 and monocyte chemoattractant protein-1 (MCP-1) levels [28]. Interestingly, it has been found that toxic liver damage promotes the recruitment of a bone marrow monocyte subset that differentiates into profibrogenic CD11b^+^F4/80^+^Gr1^+^ macrophages [29], confirming the role of extrahepatic monocytes in liver disease.

Stem cell mobilization and consecutive emigration from bone marrow is a noninvasive strategy to attract these cells for liver fibrosis treatment. It has been shown that there is a natural migration of bone marrow cells towards injured organs after transplantation [30]. Stromal derived factor 1 (SDF-1), which is upregulated in pro-inflammatory conditions, including liver damage, acts as a chemo attractant to bone marrow cells through the SDF-1/CXCR4 axis [31,32].

Several soluble factors and drugs could mobilize hematopoietic stem cells from bone marrow into the peripheral blood, such as granulocyte-colony stimulating factor (G-CSF), plerixafor, and Stem Enhance. G-CSF plus plerixafor treatment significantly improves liver function and increases liver CD34^+^ cells. G-CSF treatment also promotes hematopoietic stem cells mobilization and migration to injured liver along with TNF-α and IL-6 levels reduction. Plerixafor treatment blocks the SDF-1/CXCR4 axis, decreasing bone marrow cell migration to the liver and ameliorating liver fibrosis mainly through peroxisome proliferator-activated receptor gamma (PPARγ) and vascular endothelial growth factor (VEGF) expression increase. Stem Enhance is a natural bone marrow cell mobilizer and its therapeutic effect is associated with CD34^+^ cells increase, ECM reduction, and liver regeneration, with VEGF up-regulation and TNF-α down-regulation [33,34,35].

Experimental animal studies have shown that BMMN transplantation has beneficial effects on hepatic fibrosis, thus becoming a promising candidate for clinical trials development (Table 2). Phase 1 and 2 studies with autologous bone marrow-derived CD133^+^ and BMMN transplantation have already been completed (NCT01120925 and NCT00713934), while other BMMN clinical trials are currently in the recruiting phase (NCT03468699).

### 2.2. Endothelial Progenitor Cells

Neovascularization is fundamental to the healing of injured tissues, because it requires a supply of growth factors, nutrients, and oxygen that must be sufficiently provided by remaining and newly-formed blood vessels [40]. Studies conducted by Asahara and colleagues during the 1990s identified adult bone marrow-derived immature cells in the peripheral blood with in vitro capacity to differentiate into endothelial cells and with the ability to incorporate into sites of neovascularization, both in physiological and pathological in vivo scenarios [41,42]. These adult endothelial progenitor cells (EPCs) are rare in the peripheral blood, but can be mobilized in greater numbers from their niche in the bone marrow to the circulation by factors such as VEGF, SDF-1, G-CSF, basic fibroblast growth factor (bFGF), placental growth factor, and erythropoietin. These mobilizing factors are highly produced by peripheral tissues that undergo hypoxia during tissue damage or healing [43,44], hence recruiting circulating EPCs to the injury site by chemokine signaling [45]. Indeed, patients with liver cirrhosis have increased numbers of circulating EPCs, which correlates with hepatic disease severity [46] and with hepatic venous pressure gradient [47], suggesting that these patients have enhanced mobilization of EPCs. Furthermore, the administration of VEGF after partial hepatectomy in rats corresponded with an increase in EPC incorporation into the liver vasculature [48].

Tissue-recruited EPCs become activated to promote postnatal neovascularization and tissue repair by different mechanisms: differentiation into mature endothelial cells for de novo blood vessel formation (vasculogenesis), incorporation into injured vessels [42,49], and releasing a plethora of trophic and cyto-protective factors with paracrine effect on tissue cells. EPC paracrine action can promote angiogenesis—the process of vessel formation via pre-existing mature endothelial cells—besides tissue remodeling and regeneration, through the secretion of factors that generally include VEGF, HGF, SDF-1, insulin-like growth factor-1 (IGF-1) and MMP-9 [50,51,52,53]. Indeed, increased endogenous plasma levels of VEGF were observed after partial hepatectomy in C57BL6J mice, which correlates with the mobilization and incorporation of bone marrow-derived EPCs into regenerating liver vasculature, suggesting that EPCs became activated and secreted VEGF. Moreover, the rate of liver tissue mass regeneration after partial hepatectomy can be dependent upon EPC mobilization and incorporation into liver tissue, suggesting that these cells could be one of the regulators of liver tissue regeneration after hepatectomy [48].

The potential of EPCs to promote liver angiogenesis by regulating endothelial cell function through paracrine action becomes particularly important considering that posthepatectomy liver tissue mass regeneration is an angiogenic-dependent process directed by the regulation of endothelial proliferation and apoptosis balance [54]. It has been recently described that exosomes are active components of EPC’s paracrine role in vascular repair by up-regulating mature endothelial cell’s function [55]. Moreover, EPC-derived exosome stimulation of endothelial function is mediated by Erk1/2 signaling pathway activation [56] and miR-21-5p delivery to endothelial cells, which suppress the expression of the anti-angiogenic factor thrombospondin-1 [57].

The potential of EPCs-based therapeutic approaches to rescue liver function after tissue fibrosis has been extensively investigated in preclinical studies, showing transplanted EPC mobilization followed by incorporation into the liver parenchyma (Table 3). Transplanted bone marrow-derived EPCs proved to halt established liver fibrosis and to promote hepatic regeneration, which significantly improved survival rates through different cellular and molecular mechanisms. The described mechanisms through which EPCs can mediate liver regenerative processes in fibrotic livers include: (1) reconstitution of sinusoidal blood vessels with endothelium; (2) incorporation into hepatic sinusoids; (3) expression of growth factors, mainly VEGF, HGF, TGF-β, and EGF; (4) stimulation of liver cell proliferation; (5) suppression of activated (fibrogenic) α-SMA^+^ HSCs; (6) upregulation of endogenous TGF-β and EGF, and downregulation of TGF-β in liver cells; (7) stimulation of MMP activity by producing active forms of MMP-2, MMP-9, and MMP-13, and inhibition of liver TIMP-1 expression; (8) enhancement of eNOS (endothelial nitric oxide synthase) protein levels and upregulation of nitric oxide (NO) secretion [58,59,60,61,62].

All these antifibrogenic and growth effects mediated by EPCs result in liver regeneration through scar tissue degradation, fibrotic area reduction, recovery of hepatocyte number and vascular density, portal hypertension decrease, and hepatic blood flow improvement. EPC-secreted growth factors, like HGF, TGF-β, and EGF, could be responsible for hepatocyte proliferation. In addition, direct and indirect co-cultures of human EPCs and rat liver sinusoidal endothelial cells revealed that EPCs stimulate liver sinusoidal endothelial cells for in vitro tube formation by PDGF and VEGF secretion, demonstrating EPCs’ paracrine role in liver angiogenesis [46]. Moreover, EPCs could inhibit liver fibrosis by affecting activated HSCs. It was suggested that the inhibition of TGF-β expression by HGF, secreted from transplanted EPCs in fibrotic livers, could account for fibrosis reduction, probably by HSC apoptosis [58]. This idea is supported by studies showing that TGF-β inhibits activated HSC apoptosis [63], and that HGF inhibits liver fibrogenesis together with hepatocyte proliferation stimulation in rats with liver damage by downregulating TGF-β [64]. Co-cultures of EPCs and HSCs shed light on the mechanisms by which mobilized EPCs can modulate HSCs and reverse liver fibrosis. Liu and co-workers [65] reported that EPCs degrade the ECM, suppress both proliferation and fibrogenic activity of activated HSCs, and promote activated HSCs apoptosis in vitro through secreted cytokines. Moreover, EPCs’ ability to affect HSC fibrogenic activity and to promote apoptosis was dependent on HGF levels [65].

The above-mentioned positive results from several studies raised clinical interest in EPC-based therapies to treat liver fibrosis and improve liver function. Allogeneic strategies, however, are limited by the immunogenicity of transplanted donor cells. An attempt to overcome this limitation is the co-transplantation of mesenchymal stem cells (MSCs). The combined transplantation of MSCs with a subset of EPCs, both derived from human umbilical cord, reduced in vivo alloimmune responses to EPCs. In vitro experiments showed that inherent MSC immunosuppressive properties were responsible for reduced T-cell mediated immune responses to EPCs [66].

The effectiveness of autologous EPC-based clinical strategies depends on the number and quality of cells collected from the patient. However, altered circulating levels and function of EPCs were reported when pathological features existed, including chronic disorders [67,68,69,70]. Dysfunctional aspects include the reduction of EPC-CFU (colony forming units) number and impairments in mobilization, migratory activity, incorporation into blood vessels, differentiation, and paracrine secretion [67,70,71].

Disturbed EPCs’ commitment and differentiation potential in bone marrow-derived EPCs from mice with liver fibrosis was reported, which might be related to the clinical state of liver fibrosis in animals [72]. An enhanced in vitro proangiogenic potential of EPCs derived from cirrhotic patients compared to healthy subjects was observed [46]. It was suggested that mobilized EPCs into cirrhotic liver activate resident endothelial cells, promoting liver angiogenesis. Considering the close relationship between angiogenesis and fibrogenesis, this raises questions about the enhanced proangiogenic potential of cirrhotic EPCs, which could further aggravate fibrosis and disease progression [73,74]. Studies evaluating the relationship between cirrhotic EPCs and HSCs could clarify the role of diseased EPCs on fibrogenesis process.

In addition, Kaur and colleagues recently reported an inflammatory profile of EPCs derived from patients with alcoholic liver cirrhosis [75]. EPCs from disease patients showed lower secretion of the anti-inflammatory cytokine IL-10 and a tendency to secrete more pro-inflammatory cytokines like TNF-α and RANTES (regulated on activation, normal T cell expressed and secreted), compared to those derived from healthy controls. Altogether, these reports show important alterations in EPCs derived from patients with liver chronic damage. Impaired EPC function could lead to insufficient or excessive production of growth factors and/or differentiation, which may jeopardize liver tissue regeneration processes mediated by autologous EPCs. This hypothesis was recently challenged by Lan and colleagues using a liver fibrosis rat model [62]. They were able to isolate normal EPCs from diseased animals, which promoted hepatic neovascularization and the suppression of hepatic fibrogenesis, leading to liver regeneration and function improvement [62]. Differences in liver fibrosis induction, animal species used, and methods of EPC isolation and phenotypical identification could account for the above described discrepancies between reports. More in vitro and pre-clinical studies using human EPCs derived from liver-diseased and healthy subjects are necessary to better understand the potential clinical benefits of autologous EPC-based therapy to treat liver fibrosis.

Despite the inconsistencies regarding autologous EPC therapies, the potential of EPCs to overcome liver fibrosis and promote liver tissue regeneration inspired researchers and clinicians to conduct clinical trials using EPCs as a therapeutic agent to treat liver cirrhosis (Table 2). D’Avola and colleagues [39] reported the results of a nonrandomized, single-arm, phase 1/2 clinical trial showing that autologous bone marrow-derived EPC transplantation through the hepatic artery in patients with decompensated liver cirrhosis was feasible and safe, as no occurrence of treatment-related severe adverse events was observed up to one year follow-up. Moreover, transplanted EPCs exerted transient, but significant beneficial effects for liver function and portal hypertension. For the evaluated parameters, they concluded that in vitro expanded EPC quality was not altered by cirrhosis stage. Moreover, in vitro expanded cells presented an active EPC phenotype and produced hepato-protective growth factors like HGF and IGF-1, besides VEGF and EGF, which could underlie the potential clinical benefit of the cell therapy [39]. A Phase 3 randomized and controlled trial is now recruiting (ClinicalTrials.gov Identifier: NCT03109236) and may provide definitive data about the potential clinical benefits of EPC-based therapy to treat patients with end stage cirrhosis.

Finally, bone marrow-derived EPCs constitute a lineage and functional heterogeneous cell population, with different potential cell subsets for endothelial differentiation and for cytokine production [51,76]. Diverse subsets of EPCs were used in the above-mentioned papers, which makes it difficult to compare results and to translate them from bench to bedside. However, the future of the field of EPC-based therapy to treat liver fibrosis will benefit from recent efforts targeting the optimization and standardization of EPC definition, isolation, in vitro expansion to generate a therapeutic dose, characterization, and therapeutic potential [77,78,79,80]. Moreover, in vitro cell pre-conditioning and genetic approaches aiming to correct disease-induced cell dysfunction, enhance EPCs functions or resistance to apoptosis [81,82,83,84] will further contribute to enhance the clinical efficacy of EPC-mediated therapeutic applications for liver failure.

### 2.3. Mesenchymal Stem Cells

Friedenstein and colleagues described mesenchymal stem cells (MSCs) for the first time in the 1980s. They were defined as a cell population in bone marrow that could attach to surfaces, featuring a spindle-shaped morphology [85]. In the following years, several researchers have demonstrated that MSCs are capable of forming colonies, presenting a fast expansion rate in vitro and being capable of osteocyte, adipocyte, and chondrocyte differentiation [86,87]. Because of the lack of more specific features to define MSCs at the time, in 2006, the International Society for Cell therapy (ISCT) proposed minimal criteria to define MSCs: (1) cells must be plastic-adherent, (2) MSC population must express CD105, CD73, and CD90 and lack expression of CD45, CD34, CD14 or CD11b, CD79a, or CD19 and HLA class II; and (3) cells must be able to osteoblast, chondrocyte, and perform adipocyte differentiation. Nowadays, MSCs are recognized by their therapeutic potential [88]. Several features ensured this capacity: non-immunogenicity that allows allogenic transplantation, differentiation capacity, homing to injured sites, immunomodulatory properties, and release of molecules (growth factors and cytokines) as soluble elements or in extracellular vesicles [89,90].

In the context of liver fibrosis, several reports demonstrate the therapeutic potential of MSCs. These cells are capable of in vitro differentiation into hepatocyte-like cells upon induction with a cytokine cocktail [91], co-culture with liver cells [92], valproic acid [93], and under pellet culture condition [94]. Furthermore, in liver diseases, MSCs can exert antifibrotic effects directly by cell–cell contact or paracrine mechanism. MSC cocultivation with HSCs reduces their proliferation, inhibits their activation, and decreases α-SMA expression via the Notch1 signaling pathway activation [95] and NADPH (nicotinamide adenine dinucleotide phosphate) oxidase inhibition [96]. MSCs’ release of cytokines such as IL-10, TGF-β, and HGF can also inhibit HSC proliferation, decreasing collagen synthesis [97]. These therapeutic mechanisms can be improved by several strategies, such as 3D spheroid culture [98], priming with cytokines [99], and hypoxic conditions [100,101] that enhance antifibrotic, anti-inflammatory, and angiogenic factors produced by MSCs.

Although MSCs have confirmed antifibrotic effects in preclinical studies, there is controversy regarding this outcome. Some reports demonstrated that bone marrow derived MSCs can differentiate into HSCs and myofibroblasts, which could lead to fibrosis progression [102,103]. Ineffectiveness in improving fibrosis was also demonstrated by some authors [104,105]. Some factors, such as timing of treatment (before, during, or after liver injury) and injected cell dose, can interfere with the results. Zhao and colleagues demonstrated that antifibrotic effects on liver are more pronounced if MSCs are injected at earlier times during injury [106]. Reports that injected MSCs after injury cessation did not exhibit antifibrogenic effects [104,105]. Antifibrogenic effects also seem to be dose-dependent, as reported by Hong, who demonstrated that higher doses had a significant decrease in collagen content compared to lower doses [107]. However, these results are non-conclusive, and further studies are needed to elucidate these mechanisms.

A recent approach is the use of MSC secretome, in a strategy that can avoid some limitations involved in cell-based therapies, such as possible tumor formation [108] and loss of immune privileged status [109]. Exossomes, microvesicles, and soluble factors are released by MSCs and their therapeutic potential in liver diseases has been investigated in preclinical studies [90,110].

Currently, researchers obtain MSCs from several tissue sources beyond bone marrow, such as the umbilical cord, placenta, adipose tissue, amniotic fluid, and menstrual blood. In the following sections of this review, we are going to describe the outcomes of liver fibrosis treatment with most common sources of MSCs: bone marrow mesenchymal stem cells and adipose-derived mesenchymal stem cells.

### 2.4. Bone Marrow Mesenchymal Stem Cells

Bone marrow mesenchymal stem cells (BMMSCs) were the first isolated and identified mesenchymal cells, and therefore the most used in clinical research for the treatment of different diseases, such as liver fibrosis [111]. Among the positive effects of BMMSCs therapy are the suppression and apoptosis of HSCs, collagen reduction, liver enzyme normalization, and proinflammatory cytokine decrease. Regarding the studies of BMMSC effects on liver fibrosis, the responsible cellular and molecular mechanisms are under investigation (Table 4).

The expansion and differentiation of Th17 Interleukin-17 (IL-17) produced by CD4^+^ T cells is regulated by TGF-β1, IL-6, IL-21, and IL-23. Recent experimental studies have implicated that IL-17 signals to fibrogenesis and pro-fibrotic cytokine production in liver [112]. The neutralization of Th17 cells attenuates liver fibrosis in BDL and CCl_4_ models through IL-17, IL-1b, TNF-α, IL-2, IL-6, TGF-β serum levels reduction, and downregulation of IL17A and the IL6/STAT3 signaling pathway [113,114]. Treatment with BMMSC-conditioned medium promotes IL-10 serum level increase, produced by CD4^+^ T cells, and reduction of Th17 cells (Th17 suppression) by indoleamine 2,3-dioxygenase (IDO) release, which reduces HSC activation and collagen deposition [115]. Furthermore, BMMSC transplantation increased CD4^+^CD25^+^ Treg cells and CD161a NK cell populations in a BDL model, resulting in the suppression of inflammatory cytokines IL-1α, TNF-α, and IFN-γ, increasing IL-10, with these factors associated with M2-macrophages activation and M1-macrophages inhibition [116,117].

Recent studies demonstrated that BMMSCs transfected with hepatocyte nuclear factor 4 alpha (HNF-4α) had a better treatment effect than untreated BMMSCs on a CCl_4_ liver cirrhosis model. HNF-4α-BMMSC transplantation ameliorated liver injury, observed by an increase in albumin and cytokeratin-18 (CK-18) and decrease in aspartate aminotransferase (AST) and alanine transaminase (ALT), bilirubin total levels, and inflammation reduction through TNF-α, IFN-γ, and IL-6 decrease, and Kupffer cells inhibition. HNF-4α also promoted MSC anti-inflammatory effects, raising nitric oxide synthase (iNOS) expression, dependent on the nuclear factor-kappa B (NF-κB) signaling pathway [119,120].

It has been shown that BMMSCs improve liver regeneration by increasing HGF and MMP-2 expression and reducing CK-19 expression in a BDL model. HGF upregulation controls CK19 downregulation in the fibrotic liver, contributing to the normalization of the biliary tree [121]. Moreover, the transfection of BMMSCs with HGF potentiates the therapeutic efficacy compared to BMMSC transplantation alone and enhances stem cell migration to the injured liver [122]. In this model, HGF overexpression is related to the reduced expression of fibrogenic cytokines (PDGF-bb and TGF-β1), collagen, MMP-9, MMP-13, MMP-14, and TIMP-2, and augmentation of urokinase-type plasminogen activator [123].

Several researchers have focused on the therapeutic effects of BMMSC-derived conditioned medium (CM), extracellular vesicles, and exosomes in liver fibrosis. BMMSC transplantation, BMMSC-CM treatment, and co-culture systems have demonstrated the potential to suppress fibrogenesis, pro-inflammatory cytokine production, HSC (α-SMA) activation, and increase glycogen synthesis. BMMSC-CM promotes an increase in Th2 and Treg cells levels, reduction in Th17 cells, and AST and ALT enzymes levels [124,125,126]. Regarding treatment with BMMSCs and BMMSC-derived exosomes, both prompt fibrosis attenuation by decreasing collagen, oxidative damage parameters (by malondialdehyde and hydroxyproline), inflammatory factors (IL-1, IL-2, IL-6, IL-8, and TNF-α) and reduction in protein levels of PPARγ, Wnt3a, Wnt10b, β-catenin, WISP1, Cyclin D1, α-SMA (fibrogenic cells). However, BMMSC-derived exosomes have better effects compared with BMMSCs regarding the decrease of type I collagen, Cyclin D1 (HSCs), IL-1, and IL-6 levels [118].

Despite positive results related to BMMSC therapy, some studies have demonstrated opposite results. Several studies have shown that BMMSCs could contribute to HSC and myofibroblast populations, leading to the maintenance of fibrogenic areas [127]. A study showed that in female mice that received lethal irradiation and bone marrow transplant from male mice, followed by CCl_4_ or thioacetamide induction of cirrhosis, most hepatic stellate cells and myofibroblasts in the liver were confirmed to have origins in the BM, and were responsible for fibrosis development [103]. In a similar way, another group found that sublethally irradiated NOD/SCID (Nonobese diabetic/severe combined immunodeficiency) mice that received single or chronic administration of CCl_4_, when transplanted with human BMMSCs, did not show any improvement in hepatic functions. In the chronic CCl_4_ model, myofibroblasts in the fibrotic animals appeared to be mainly of human origin [128]. In both of the previous studies, BMMSC did not contribute to significant amounts of parenchymal cells. Possible mechanisms underlying BM contribution to fibrogenic cells may point to phingosine 1-phosphate (S1P), a bioactive lysophospholipid upregulated in liver fibrosis. S1P mediates the homing of BMMSCs (by S1P3 receptor) and could influence its differentiation into myofibroblasts [103,129].

Regarding BMMSC transplantation clinical trials (Table 2), currently there are phase 1 and 2 studies in the recruitment stage (NCT03838250 and NCT00993941, respectively), and a phase 3 study in an unknown situation (NCT01854125). The NCT01854125 clinical trial had positive results in phases 1 and 2, demonstrating liver improvement, confirmed by the MELD (Model for End-Stage Liver Disease) score [38].

### 2.5. Adipose-Derived Mesenchymal Stem Cells

Adipose-derived mesenchymal stem cells (ADSCs) are a promising tool in regenerative medicine. They can be easily obtained from liposuction procedures and propagated ex vivo [130]. Experimental approaches with ADSCs demonstrated enthusiastic results in models of several diseases, such as ischemic injuries [131], myocardium infarction [132], and autoimmune disorders [133].

Comparatives studies between BMMSCs and adipose tissue-derived mesenchymal stem cells (ADSCs) demonstrate that both sources are effective in liver fibrosis treatment, but ADSCs promote a higher inhibition of proliferation, activation, and apoptosis of HSCs, in addition to a better AST/ALT ratio. These results, combined with the facility of obtaining these cells from adipose tissue compared to bone marrow, suggest that ADSCs could be a great alternative source for therapy [134,135].

Regarding liver fibrosis, several preclinical studies demonstrated that ADSCs can attenuate liver fibrosis, as indicated in Table 5. The mechanisms of ADSC-mediated fibrosis resolution include HSC inhibition [91,136], the release of angiogenic and anti-inflammatory factors [137], antioxidant pathways upregulation [138], MMP increase [91], and apoptotic pathway modulation [139].

A recent report demonstrated that fibrotic rats who received ADSCs had higher serum levels of anti-inflammatory factors (HGF and IL-10) and diminished inflammatory cytokines (TGF-β and TNF-α). Besides this, liver collagen deposition was decreased and proliferating cell nuclear antigen (PCNA) was increased, with fibrosis improvement and hepatocyte proliferation. However, the mechanism of fibrosis resolution was not completely elucidated [140].

Hepatocyte-like cells derived from ADSCs can be transplanted into fibrotic livers. Heterologous transplant of pig ADSCs-differentiated hepatocytes was used in a mouse model of liver failure. These cells were able to decrease the serum levels of ALT, AST, and TP, and increase albumin levels three weeks after transplantation. Cells could migrate to the liver and collagen deposition was decreased. However, cell fusion of transplanted cells with mouse hepatocytes was not evident and the mechanism of liver fibrosis resolution was unknown [142]. In other research, human ADSC-derived hepatocytes transplanted in a mouse model of liver fibrosis demonstrated improved serum levels of liver injury biochemical markers. However, non-differentiated ADSCs exhibited better results with a more accentuated decrease of ALT and AST three weeks after transplantation in comparison to ADSC-derived hepatocytes [143]. A divergent study showed that liver-injured rats who received ADSC-derived hepatocytes had better levels of serological markers (albumin, ALT, bilirubin, and alkaline phosphatase), increased levels of hepatic genes (albumin, alpha-fetoprotein, cytokeratin18, and HNF), and better cell homing than liver-injured rats who received undifferentiated ADSCs [141]. Regarding liver fibrosis, collagen deposition and α-SMA was quantified and both differentiated and undifferentiated ADSCs were capable of decreasing liver fibrosis, with no significant difference between these cells. It is noteworthy that the serum of fibrotic rats was used to differentiate ADSCs towards hepatic lineage. This serum had increased levels of VEGF, SDF1, and EGF, factors that can boost the therapeutic potential of ADSCs [141].

ADSCs can also have their therapeutic potential in liver fibrosis enhanced by preconditioning with cytokines [91], polyphenols [138], 3D culture [98], transfection with plasmids (FGF-2) [145], HGF, and miRNA [110].

ADSCs treated with a cocktail of cytokines (VEGF-A, EGF, IGF, and FGF) exhibited upregulated hepatogenic (CK18, alpha-fetoprotein, albumin) and chemotactic genes (IGFBP, HGF, G-CSF). An improvement in liver fibrosis in mice after transplantation of preconditioned ADSCs was evident, showing decreased collagen content and α-SMA, increased liver angiogenesis, hepatocyte proliferation, and MMP-13 production. Also, conditioned medium from stimulated ADSCs inhibited HSC activation in vitro, with decreased expression of α-SMA [91].

Resveratrol preconditioned ADSCs could improve biochemical parameters, decrease collagen content, and activate antioxidant pathways in a liver injury model induced by diabetes. Resveratrol is a polyphenol recognized by its antioxidant properties. After transplantation with these conditioned cells, the liver exhibited increased Sirt1 and SOD 2 (related to antioxidant signaling) and MMP-2 [138].

Conditioned medium of ADSCs transfected with FGF-2 inhibited signaling pathways of p-JNK, NF-κB and p-Smad2/3 (involved in expression of α-SMA and collagen) in LX-2 cell activated by TGF-β in vitro. This inhibition was associated with the increased expression of α-lactoalbumin and lactotransferrin in the secretome of these transfected cells compared with empty ADSCs. In vivo, liver fibrosis was also improved with FGF2-transfected ADSCs transplantation in thioacetamide-induced hepatic fibrosis. Decreased levels of serum hyaluronic acid and reduced expression of fibrosis-related factors such as α-SMA, collagen, and TIMP-1 were observed [145].

Cultivation of ADSCs in a 3D environment increased the expression of HGF, IGF-1, and IL-6 compared to a 2D environment. Conditioned medium of 3D ADSCs decreased apoptosis, LDH (lactate dehydrogenase) release, Bax/Bcl-2 ratio, NF-kb, and TGF-β in injured hepatocytes in vitro. After spheroids transplantation in a mice model of hepatic fibrosis, liver function was improved, demonstrated by biochemical marker serum levels and decreased type I and III collagen in hepatic tissue [98].

Regarding clinical application of these cells, currently there are seven studies registered at clinicaltrials.gov, as demonstrated in Table 2 (NCT02297867; NCT02705742; NCT04088058; NCT03629015; NCT00913289; NCT01062750; NCT03254758). The only clinical trial that has published results demonstrated that the application of freshly isolated ADSCs was safe and no serious adverse events were reported. Patients maintained or improved albumin serum levels during follow-up [37].

Although ADSCs are recognized as an allied tool in regenerative liver medicine, there is still controversy regarding the route of administration, timing of transplantation, and outcomes of short- and long-term use. Therefore, ADSCs have great potential to treat liver injuries, but further studies are needed to better understand all the mechanisms involved in liver regeneration.

### 2.6. Primary and Cell Line Hepatocytes

The replicative nature of hepatocytes has long been explored as a favorable condition for in vitro expansion, but conventional bidimensional culture systems lack the signaling milieu necessary for maintaining primary hepatocytes outside the liver, resulting in a loss of many physiological functions and normal morphology. Well-established hepatocyte cell lines like HepG2 (derived from hepatoma), although extensively characterized, do not express key hepatic enzymes in comparable physiological levels, and accumulate several features of transformed cells. The recently established HepaRG line, derived from human hepatoma, is originally formed from bipotential progenitors that give rise to both hepatocytes and bile canaliculi. When differentiated into mature hepatocytes, HepaRG cells display similar gene expression and consistent hepatocyte morphofunctional features, with granular appearance and growing in clusters known as “hepatocyte islands”. HepaRG cells also present iron-loading capacity, induction of CYP (cytochrome P450) proteins, and the hepatocyte derivatives are sensible to acetaminophen injury, in a similar manner to primary hepatocytes [146]. However, these cells lack the typical genetic heterogeneity of human hepatocytes, such as variable ploidy and gene expression, which have been implied in hepatocyte zonation within the hepatic lobule and are proposed to be advantageous features for liver metabolism and physiological turnover [147,148].

In the past decade, several 3D scaffolds and bioreactors systems have been used to maintain human primary hepatocytes in near-physiological conditions [149]. However, the several ECM molecules embedded within these scaffolds have been shown to confine drugs and metabolites, disfavoring the ideal conditions for hepatocyte metabolism and expansion. Problems with reproducibility and larger scale production have also been reported [150]. The most successful attempts to cultivate these cells have been achieved using the spheroid model of 3D culture [151]. Using this approach, Bell and colleagues [152] showed that primary human hepatocytes spheroids can be maintained up to five weeks in serum-free and well-defined media, displaying a typical secretome profile, morphology, and physiology, confirmed by albumin and urea secretion, gene expression, and activity levels of CYP complex enzymes comparable to human healthy liver tissue. Importantly, these cells would integrate with other non-parenchymal cells, such as cholangiocytes, Kupffer cells, and HSCs. This implies that diseases like cholestasis and steatosis could be successfully modeled in this microtissue model, allowing robust drug-screening analysis.

Regarding the effectiveness of lineage or primary hepatocytes to treat liver fibrosis, animal models have provided better overall results compared to the effects observed in humans. The transplantation of primary hepatocytes extended survival in Nagase analbuminemic rats subjected to BDL, increasing albumin and bilirubin levels [153]. Cai et al. [154] showed that the transplantation of alginate-encapsulated immortalized hepatocytes had therapeutic properties similar to primary hepatocytes in a model of hyperammonemia-induced hepatic encephalopathy, improving survival, prothrombin time, serum albumin, and bilirubin levels. Interestingly, human primary hepatocytes co-transplanted with HSCs into SCID mice with healthy livers exhibited a better engraftment capacity than hepatocytes transplanted alone, and HSCs did not seem to contribute to liver fibrogenesis in this model [155].

Usually, past clinical studies using primary hepatocytes focused on metabolic syndromes such as factor VII deficiency, Crigler-Najjar syndrome type 1 (CN-1), and urea cycle defects. Patients that received these cells experienced several months of disease regression, but usually orthotopic liver transplantation would still be required [156]. The most recent ongoing clinical trial registered on hepatocytes derivatives is a Phase I study based on the use of a bio artificial liver tissue obtained from the in vitro manufacturing of autologous liver resections from patients with end-stage hepatic disease. The product, a hepatocyte matrix implant, is made of self-dissolving polymers cultivated with human autologous hepatocytes (ClinicalTrials.gov Identifier: NCT01335568).

Despite all the advances in the development of ideal culture systems for non-autologous primary hepatocytes, these cells still are subject to up to 70% of immune rejection from the patient´s organism and viability loss following cryopreservation/thawing cycles [157]. These factors may result in poor engraftment capacity when transplanted into injured livers, which is a major barrier for the use of primary hepatocytes for cell therapy.

### 2.7. Oval or Hepatic Progenitor Cells

In the past decades, bipotential populations of liver progenitor/stem cells have been described, and although these cells most likely represent a heterogeneous population, they have been collectively named oval cells or hepatic progenitor cells. They have been implicated in the ductular reaction that is observed after certain liver injuries, and is defined as a process driven by these intermediate ductular cells, mainly located in the Hering ductules, which actively proliferate and form new bile ducts, being associated with liver fibrosis [158]. Additionally, oval cells are especially activated when hepatocyte and/or duct cell proliferation are hampered and can differentiate into both cell types, which makes these cells eligible candidates for cell therapy in liver diseases. During this differentiation process, sometimes oval cells share the expression of common markers between both parenchymal liver populations, such as AFP (alpha-fetoprotein), albumin, and cytokeratins CK-8, CK-18, and CK-19 [159].

Recent studies have shown that in some situations, such as after liver orthotopic transplantation in small-for-size fatty liver grafts, oval cells may undergo mesenchymal transition and contribute to liver fibrosis [160]. In a BDL model of cholestasis, four populations of ultrastructurally distinct hepatic progenitor/oval cells were described. In addition, evidence shows the activation and proliferation of a ductular population, which makes contact with macrophages, HSCs, and portal cells, contributing to liver fibrogenesis in this model [161].

However, most studies confirmed that oval cells are often engaged after chronic liver injury, especially in advanced fibrosis and cirrhosis, and contribute to liver regeneration, providing new parenchymal cells. Although this observation has prompted some research for the use of oval cells as therapeutic agents, the difficulty in isolation and in vitro expansion of respective cells and their overall low prevalence has restrained further therapeutic advances in this field. Oval cells also express MSC markers (c-kit and Thy-1 or CD90), and recent work has shown that murine MSCs can differentiate into oval cells that contribute to liver regeneration in a CCl_4_-induced fibrosis model [162]. Also, in a cholestatic model of a DDC-diet in bone morphogenetic protein 9 (BMP9)-deficient mice, BMP9 was shown to negatively regulate oval cell activation and ductular reaction, reducing liver regeneration [10].

All these findings point to the complex nature of oval/hepatic progenitor activation mechanisms and its role in liver fibrosis or regeneration, reassuring the importance of studying molecular events regarding these cells during liver disease and therapy [163].

### 2.8. Pluripotent Cells

In liver development, progenitor bipotential endodermal cells, known as hepatoblasts are defined as the precursors of both hepatocytes and cholangiocytes. When next to a portal vein, these hepatoblasts form an early epithelial biliar plate, which gives rise to the tubular biliary tree, in a process highly dependent on the Notch2 gene and Notch2/Jagged1 pathway. Other factors, such as TGF-β, EGF, HGF, Hedgehog (Hh), and Wnt, are required [164].

The generation of functional hepatocytes from stem cells has been a challenge [15], because they have extremely specialized functions, complex morphophysiology, play a myriad of metabolic and secretory roles, and establish a rich interconnected net with other liver parenchymal cells (endothelial cells, HSCs, cholangiocytes, Kupffer cells, and immune cells). In the past years, most attempts to differentiate hepatocytes from pluripotent cells have achieved the so-called hepatocyte-like cells, which designates a differentiated cell that has most hepatocyte properties, but usually lacks some features regarding gene expression profile, protein expression, and/or metabolic activity [165].

To obtain hepatocytes for future patient-specific protocols, induced pluripotent cells (iPSC) obtained from somatic cells after triggering the expression of the classical “Yamanaka” factors (OCT4, SOX2, KLF4, and c-MYC), or other factors/techniques have arisen as excellent candidates. However, as with any pluripotent stem cell-derived sample, a major safety issue is the residual teratogenic/tumorigenic activity in differentiated cells, which is associated with latent pluripotency [14,15,166]. Despite these limitations, these cells can still be valuable for therapeutic use, once they prove to be effective and safe. To date, there are no registered clinical trials ongoing with hepatocyte or hepatocyte-like cells derived from pluripotent stem cells (PSCs) differentiation. Most studies show the successful derivation of human hepatocyte-like cells from diverse types of PSCs (embryonic and iPSCs), forecasting the need to avoid feeder cell layers or exogen products (serum, Matrigel) [166,167]. Table 6 summarizes some relevant and promising pre-clinical results obtained from pluripotent stem cells in different models of liver fibrosis.

In 2017, Takayama and collaborators reprogrammed peripheral mononuclear blood cells into HLA-homozigous and non-HLA-homozigous human induced pluripotent stem cells (iPSCs) using the plasmids pCE-hSK, pCE-hUL, pCE-hOCT4, and pCE-mp53DD [171]. These cells attached to a mixture of different fragments of human recombinant laminin better than to collagen, and after endoderm specification, generated hepatoblasts expressing alphafetoprotein (AFP) and HNF-4a (hepatocyte nuclear factor 4a). Particularly, the cultivation of these cells on the laminin fragment LN111 resulted in minimal residual pluripotency, analyzed with the Tral-60 marker. Incubation of hepatoblasts with oncostatin M and HGF over a collagen type IV matrix retrieved hepatocyte-like cells with expression levels of albumin and asialoglycoprotein receptor-1 comparable to primary human hepatocytes, as well as functional cytochrome P450 activity.

The cells and molecules behind cholestatic fibrosis, which is the result of any chronic obstruction in the bile flow within intrahepatic and/or extrahepatic bile duct systems, are poorly understood, in part due to the high difficulty level of disease-modeling cholangiopathies ex vivo. Most liver organoids or biologic systems engineered to date lack a functional or recognizable bile duct system, and the poor bile draining out of these organoids can lead to tissue injury and malfunctioning, with these systems not being suitable for future clinical use. The creation of epithelial cells similar to cholangiocytes from bile ducts is possible in 2D and 3D models, using an EpCAM-enriched population of hepatic spheroids derived from iPSCs [172]. Other studies differentiated cholangiocytes by cultivating hepatoblasts with FGF-10 (fibroblast growth factor-10), activin, and retinoic acid and exposing cells to a 3D organoid structure [173]. However, the applicability of these in vitro/pluripotent cells-derived cholangiocytes in liver disease has not yet been tested in cholestatic patients, and more data concerning the integration of bioengineered bile duct systems within functional liver tissue are needed.

Alternatively, considering the liver in the context of the immune system, it has been shown that at least a part of the hepatic immune niche can be reconstituted from pluripotent cells. Tasnim and colleagues (2019) [174] recently achieved the in vitro generation of functional Kupffer cells, with typical phenotype and inflammatory activity. This provided a model that allows a deepened study on the immune interactions in hepatic diseases, hepatic drug metabolism, and hepatotoxicity analyses, especially in co-culture conditions adding hepatocytes and cholangiocytes. Furthermore, the notion that primitive erythro-myeloid progenitors from the yolk sac are the source of primitive macrophages that give rise to differentiated Kupffer cells, and not blood circulating monocytes has favored the development of new in vitro protocols for obtaining these non-parenchymal cells [175].

## 3. Conclusions

The development of novel cell therapy strategies for end-stage liver disease, which is the consequence of uncompensated fibrosis/cirrhosis, is in constant evolution, prompted by the developing nature of biotechnology processes and by the unveiling of several molecular aspects and cell interactions within the hepatic niche in health and disease. New discoveries on liver embryogenesis and adult liver regenerative mechanisms are continuously providing hints and pointing out potential targets for advanced treatments. These may combine several cell sources or products, such as secretome molecules and/or matrix scaffolds, along with modern state-of-the-art surgical and pharmacological approaches in personalized protocols for each patient, disease, and respective staging.

In the last two decades, multiple cell therapy techniques developed to treat liver fibrosis/cirrhosis gathered knowledge about in vitro cell manipulation improvement and the mechanisms by which transplanted cells could reverse liver fibrosis and promote liver regeneration. This knowledge is now supporting the development of new approaches targeting the generation of in vitro systems that could lead to the development of liver organoids or even whole bioengineered livers for transplantation, aided by emerging sophisticated technologies like 3D (bio)printing, microfluidics, and bioreactors, as recently reviewed elsewhere [176].

Finally, cell therapy is a promising tool among the current Medicine techniques to treat liver diseases and may be used in accordance to provide the best match for treatment, in all efforts to avoid the frequently dire outcome of the need for liver transplantation.

## Figures and Tables

**Figure 1 cells-08-01339-f001:**
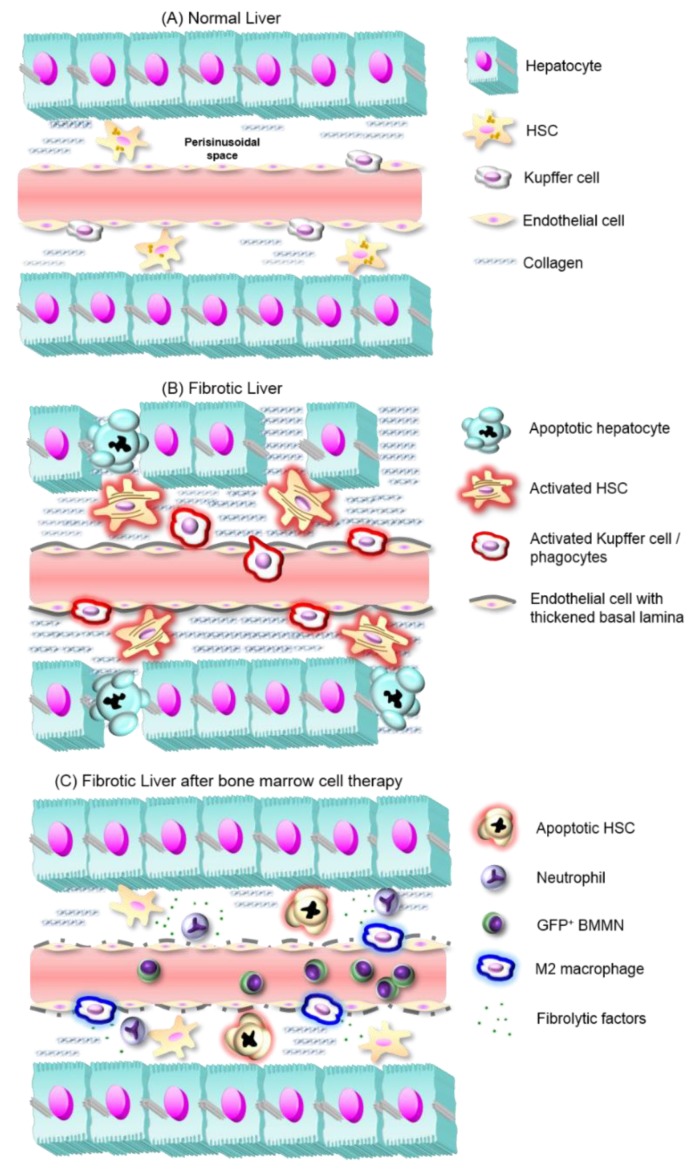
Liver fibrosis development and proposed mechanisms of fibrosis reversal by BMMNs. (**A**) In normal liver, hepatocyte plates line sinusoidal capillaries and between both structures the submicroscopic perisinusoidal (also subendothelial or Disse space) homes hepatic stellate cells (HSCs), which produce small amounts of a reticular extracellular matrix (ECM), and stock retinyl esters. (**B**) In fibrotic liver, activated Kupffer cells and other inflammatory cells release proinflammatory cytokines, which in turn activate HSCs. These cells increase the deposit of a fibrous ECM in perisinusoidal space. Along with de novo synthesis of a thickened basal lamina around hepatic sinusoids, these events cause hepatocyte hypoxia and apoptosis. (**C**) After BMMNs expressing green fluorescent protein (GFP^+^ BMMNs) transplantation, Kupffer cells quantities decrease, while BMMC-derived neutrophils and macrophages with an anti-fibrotic phenotype appear and release anti-inflammatory cytokines. These factors contribute to HSC quiescence and/or apoptosis, and tissue remodeling that ensures the regeneration of liver function.

**Table 1 cells-08-01339-t001:** Experimental liver fibrosis and bone marrow mononuclear cells (BMMNs).

Cells/Dose/Route	Fibrosis Model	Main Results and Mechanisms of Action	Reference
Rat BMMNs, 10^7^ cells, jugular vein	BDL (Wistar rats)	Collagen types I and IV, laminin, CK-19 and α-SMA reduction	[22]
Rat BMMNs, 10^7^ cells, jugular vein	BDL (Wistar rats)	MMP-9 and MMP-13 expression were increased by macrophages, TIMP-1 and TIMP-2 reduction	[23]
Rat BMMNs, 10^7^ cells, jugular vein	BDL (Wistar rats)	Fibrogenic cell apoptosis	[24]
Rat BMMNs, 10^7^ cells, jugular vein	BDL (Wistar rats)	Oxidative stress reduction (4HNE), mitochondrial coupling (UCP2 levels) and biogenesis (PGC1-α) regulation	[25]
Mouse BMMNs, 10^7^ cells, jugular vein	BDL (C57BL/6)	BMMNs originated populations of CD144, CD11b and Ly6G cells in the fibrotic liver, anti-fibrotic cytokines augmentation (IL-10, IL-13, IFN-γ, HGF) and pro-inflammatory cytokines reduction (IL-17A, IL-6)	[26]
Mouse BMMNs or BMMNs-derived monocytes, 10^6^ cells per 3 weeks	CCl_4_ (orogastric), 200 μL-20%, 12 weeks, C57BL/6	BMMN derived-monocyte had a better therapeutic effect, pro-inflammatory/fibrotic cytokines (TNF-α, IL-6, IL-1β, TGF-β1) reduction, IL-10 and MMP-9 were increased	[27]
Mouse BMMNs, 10^6^ cells, tail vein	CCl_4_ (intraperitoneal) 0.4 mL/kg, 3 x per week, 2 weeks, C57BL/6	CD4^+^CD25^+^Foxp3^+^ Treg cells produced IL-10 and promoted IL-6 and MCP-1 reduction, CD11b^+^F4/80^+^ cells were reduced in fibrotic liver	[28]

4HNE: 4-Hydroxynonenal, BDL: bile duct ligation.

**Table 2 cells-08-01339-t002:** Clinical trials using cells as therapeutic agents for inhibition of liver fibrosis/cirrhosis, registered under ClinicalTrials.gov [36].

Trial Number (Status)	Cohort	Intervention	Study Phase Type	Follow-Up (Months)	Main Analysis Criteria	Outcomes/Published Results
NCT02297867 (Completed)	Liver cirrhosis (n = 6)	Autologous ADSC by intrahepatic injection	Phase I	1–6	MELD	NR
NCT02705742 (unknown)	Liver cirrhosis	Autologous ADSC by intravenous injection	Phase I/II	12	All-cause mortality	NR
NCT04088058 (not yet recruiting)	Liver cirrhosis (n = 20)	Autologous ADSC by intrahepatic injection	Phase II open-label single-arm	1–12	MELD	NR
NCT03629015 (not yet recruiting)	Acute liver failure (n = 20)	Allogeneic ADSC by intravenous infusion of low (0.5 × 10^6^ cells/kg) or high (2 × 10^6^ cells/kg) dose	Phase I	12	Incidence of adverse events and suspected unexpected serious adverse reaction	NR
NCT00913289 (terminated)	Liver cirrhosis (n = 6)	Autologous adipose tissue-derived stromal cells	Phase I	6	All cause harmful events	NR
NCT01062750 (Completed)	Liver cirrhosis (n = 4)	Autologous adipose tissue-derived stromal cells via intrahepatic arterial catheterization	NA	1	All cause harmful events	No serious adverse events, albumin serum levels were improved in three patients [37]
NCT03254758 (Recruiting)	Decompensated liver cirrhosis	ADSC by intravenous infusion	Phase I/II	6	Incidence of adverse events and Child Pugh score	NR
NCT01854125 (unknown)	Liver cirrhosis (n = 30)	Autologous BMMSC transplantation via hepatic artery catheterization	Phase III	3	LF, MELD, adverse effects	Improvement of LF in cirrhotic patients after autologous mesenchymal stem cell injection in phase I–II [38]
NCT00993941 (unknown)	Liver cirrhosis	Autologous BMMSC transplantation via portal vein catheterization or drug therapy (oral or intravenous)	Phase II	12	ALT, total bilirubin, prothrombin time, albumin, laminin, prealbumin, procollagen III, collagen IV, hyaluronidase and histology	NR
NCT03838250 (Recruiting)	Alcoholic liver cirrhosis (n = 10)	Autologous BMMSC transplantation via hepatic artery	Phase I	12	Incidence of serious adverse events	NR
NCT03468699 (Recruiting)	Biliary liver cirrhosis (n = 20)	Autologous BMMSC transplantation via hepatic artery	Single group assignment, Phase II	6	Cholestasis changes, LF, PELD	NR
NCT00713934 (Completed)	Liver cirrhosis (n = 7)	Autologous BM-derived CD133^+^ and BM mononuclear stem cells transplantation via portal vein	Randomized Phase I/II	6	LF, MELD	NR
NCT01120925 (Completed)	Liver cirrhosis (n = 30)	Autologous BM-derived CD133^+^ and BM mononuclear stem cells transplantation via portal vein	Randomized Phase I/II	6	LF, MELD and Child Pugh scores	NR
NCT01333228 (Completed)	Liver cirrhosis (n = 12), age 18–75 years	Autologous BM-derived EPCs, single group assignment, 8.45 × 10^6^ to 450 × 10^6^ cells administered through the hepatic artery	Single arm non-randomized Phase I/II	12	Primary: Number of Participants with adverse events; Secondary: LF, MELD, and Child-Pugh scores, HVPG, complications of liver cirrhosis	Treatment was confirmed safe and feasible, transient (but significant) beneficial effects in LF [39]
NCT03109236 (Recruiting)	Decompensated liver cirrhosis (n = 66)	Autologous BM-derived EPCs administrated via the portal vein system	Two arm randomized Phase III	3, 6, or 12	Primary: Fibrosis (Ishak, MRE, MELD, quantitative fibrosis), Secondary: overall survival, LF, HVPG, clinical decompensation, patient reported outcome	NR

ALT: alanine aminotransferase, ADSC, adipose-derived stem cell(s), BM: bone marrow, BMMSC: Bone marrow mesenchymal stem cell(s), EPC: Endothelial progenitor cell(s), HVPG: hepatic venous pressure gradient, LF, liver function, MELD: Model for End-stage Liver Disease, MRE: Magnetic resonance elastography, NA: not applicable, NR: no results available yet, PELD: Pediatric end-stage liver disease.

**Table 3 cells-08-01339-t003:** Experimental liver fibrosis and endothelial progenitor cells.

Cells/Dose/Route	Fibrosis Model	Main Results and Mechanisms of Action	Reference
Rat BM-EPCs,3 × 10^6^ cells—single or four-repeated doses once a week for 4 weeks, tail vein	CCl_4_ or thioacetamide (intraperitoneal) twice a week for 10 weeks (Wistar)	Increased survival rates, liver fibrosis and fibrogenesis reduction (HSC suppression and enhanced MMP activity), increased hepatocyte proliferation and HGF, TGF-α, EGF, and VEGF expression in liver	[58]
Rat BM-EPCs, 5 × 10^5^ cells, portal vein	CCl_4_ by gavage twice a week for 16 weeks (Sprague-Dawley)	Increased survival rates, reduced levels of AST, ALT, and TBIL, albumin levels restoration, liver fibrosis and fibrogenesis reduction (HSC suppression), increased liver cell proliferation	[59]
Rat BM-EPCs, 3 × 10^6^ cells—once weekly for four weeks, tail vein	Dimethylnitrosamine (intraperitoneal) three times a week for eight weeks (Sprague–Dawley)	Liver fibrosis and fibrogenesis reduction (HSC suppression), increased hepatocyte proliferation, vascular density and HGF, TGF-α and EGF expression in liver	[60]
Rat BM-EPCs, 3 × 10^6^ cells—once a week for four weeks, tail vein	CCl_4_ (intraperitoneal) twice weekly for 10 weeks (Wistar)	Liver fibrosis and fibrogenesis reduction (HSC suppression), reduced portal venous pressure, increased vascular density and hepatic blood flow	[61]
Fibrotic rat BM-EPCs, 2 × 10^5^ and 2 × 10^6^ cells—once a week for three weeks, tail vein and portal vein	CCl_4_ (subcutaneous) twice a week for six weeks (Wistar)	Liver fibrosis suppression, improved liver function (lower ALT, AST, APTT), increased liver mRNA levels of HGF and VEGF, increased liver cells proliferation	[62]

BM-EPCs: Bone marrow-derived endothelial progenitor cells; HSC: hepatic stellate cells; MMP: matrix metalloproteinase; HGF: hepatocyte growth factor; TGF-α: transforming growth factor-α, EGF: epidermal growth factor; VEGF: vascular endothelial growth factor; AST: aspartate aminotransferase; ALT: alanine aminotransferase; TBIL: total bilirubin; APTT: activated partial thromboplastin time.

**Table 4 cells-08-01339-t004:** Experimental liver fibrosis and bone marrow mesenchymal stem cells.

Cells/Dose/Route	Fibrosis Model	Main Results and Mechanisms of Action	Reference
Mouse BMMSC and CM-BMMSC, 10^6^ cells	CCl_4_ (intraperitoneal) in C57BL/6 mice, 1 µL/g, 2x per week, 1 month	IDO promoted Th17 suppression (IL-17 reduction) and IL-10 production and activation of CD4 T cells.	[115]
Human BMMSC and EX-BMMSC, 10^6^ cells	CCl_4_ (intraperitoneal) in Sprague Dawley rats	EX-BMMSC had better therapeutic effect; IL-1, IL-2, IL-6, IL-8, and TNF-α reduction; PPARγ, Wnt3a, Wnt10b, β-catenin, WISP1, Cyclin D1 were decreased; inhibition of hepatic stellate cell activation	[118]
Mouse BMMSC and HNF-4α-overexpressing BMMSC, 10^6^ cells	CCl_4_ (intragastric gavage) in C57BL/6 mice, 5.0 mL/kg, 2x per week, 3 weeks	HNF-4α-overexpressing BMMSC had better therapeutic effect; reduction in TNF-α, IFN-γ, IL-6; enhanced iNOS expression that depends on NF-κB signalling; Kupffer cell inhibition	[119]
Rat BMMSC, 3 × 10^6^ cells	CCl_4_ (intraperitoneal) in Sprague Dawley rats, 1 mL/g, 2x per week, 6 weeks	Reduction in IL-17, IL-2 and IL-6; downregulation of IL-17a, IL-17ra, IL-17f, Stat3, p-STAT3, Stat5a, p-SMAD3, and TGFβR2; elevation of p-STAT5 protein	[114]
Rat BMMSC, HGF overexpressing BMMSC and HGF alone	CCl_4_ (intraperitoneal) in Sprague Dawley rats, 5.0 mL/kg, 30 consecutive days	AST, ALT, total bilirubin levels reduction; hepatocyte nuclear factor 4α, albumin, and cytokeratin 18 expression were increased	[120]
Mouse BMMSC, 5 × 10^5^ cells	CCl_4_ (intraperitoneal) in C57BL/6 mice, 1 mL/g, 70 days	IL-10 increase and IL-12b, IFN-γ, TNF-α, and IL-6 gene expression were decreased; M2 macrophages activation; MMP13 production and M1 macrophages inhibition	[117]
Rat BMMSC, 10^6^ cells	Bile duct ligation in Sprague Dawley rats	T cell proliferation was decreased; inflammatory cytokines were reduced; expansion of intrahepatic NK cells	[116]
Rat BMMSC, 10^6^ cells or Silymarin (100 mg/kg)	Bile duct ligation in Wistar rats	BMMSC had a better therapeutic effect; MMP-2 mRNA upregulation and CK-19 mRNA downregulation; HGF augmentation regulate MMP-2 and CK-19 gene expression	[121]
Human BMMSC, HGF-overexpressing BMMSC, 4 × 10^6^ cells	CCl_4_ (intraperitoneal) in Sprague Dawley rats, 0.2 mL/100 mg, 3x per week, 8 weeks	HGF overexpressing BMMSC had better therapeutic effect; HGF overexpressing increased homing of BMMSC in the liver	[122]
Human BMMSC, HGF-overexpressing BMMSC, 10^7^ cells	Dimethylnitrosamine (intraperitoneal) in Sprague Dawley rats, 1 mL/g, three consecutive days per week, 4 weeks	HGF overexpressing BMMSC had better therapeutic effect; reduction in PDGF-bb, TGF-β1, and TIMP2; inhibition of α-SMA cells	[123]

**Table 5 cells-08-01339-t005:** Experimental liver fibrosis and adipose-derived mesenchymal stem cells.

Cells/Dose/Route	Fibrosis Model	Main Results and Mechanisms of Action	Reference
Human ADSC induced with cocktail of cytokines,1 × 10^6^ cells by intravenous route	thioacetamide (intraperitoneal) in BALB/c nude mice (150 mg/kg) twice a week for 1 month	Degradation of MEC (increased MMP-13), decreased collagen and α-SMA content, activation of p-38 MAPK signaling, angiogenesis and hepatocyte proliferation, inhibition of HSC	[91]
Rat ADSC preconditioned with resveratrol, 1 × 10^6^ cells by intravenous route	Streptozotocin (intraperitoneal) in Wistar rats (50 mg/kg)	Decreased collagen I, improved hepatocyte survival, downregulation of apoptotic pathway (caspase-3, cytochrome-c, FAD), upregulation of antioxidant pathway (Sirt1, SOD2), degradation of MEC (increased MMP-2)	[138]
Rat ADSC preconditioned with serum of liver- injured rat, 1.5 × 10^6^ cells by intrahepatic route	CCl_4_ (intraperitoneal) in Sprague Dawley rats, 1 µL/g twice a week for one month	Decreased fibrosis, increased albumin, AFP, CK-18 and HNF4 levels in liver, degradation of MEC (increased MMP-2), increased expression of CK-8, CK-19, albumin, and AFP in ADSC	[141]
Rat ADSC, 3.0 x10^6^ cells	CCl_4_ (intraperitoneal) in Wistar rats, three times a week for two months	Increased HGF and IL-10 serum level, decreased TGF-β and TNF-α, decreased collagen deposition, increased PCNA, hepatocyte proliferation	[140]
Pig ADSC induced to hepatocytes, 1.5 × 10^7^ cells by intravenous route	CCl_4_ (intraperitoneal) in mice, twice a week for five to seven weeks	Decreased collagen deposition, decreased ALT and AST, increased albumin serum levels, decreased TGF-β, IL-6 and IL-10	[142]
Human ADSC induced to hepatocytes, 2–3 × 10^6^ cells by intravenous route	CCl_4_ (intraperitoneal) in mice, 5 mL/kg single dose and Cyclosporin A (10 mg/kg) daily for three weeks	Decreased ALT and AST	[143]
Rat ADSC preconditioned with bFGF, 5.0 × 10^6^ cells by intravenous route	CCl_4_ (intraperitoneal) in Fischer 344 rats, 1 mg/kg twice a week for eight weeks	Apoptosis of HSC, activation of JNK-p53 signaling, increased expression of HGF	[139]
Human ADSC pretreated with Lysophosphatidic acid and sphingosine-1-phosphate, 2 × 10^6^ cells by intravenous route	Intraperitoneal D-galactosamine (600 mg/kg) and LPS (8 μg/kg) in NOD/SCID mice	Decreased serum levels of ALT, AST, MDA, TNF and caspase-3/7	[144]
Rat ADSC, 5.0 × 10^6^ cells injected through portal vein	CCl_4_ (subcutaneous) in Sprague Dawley rats, 1.5 mL/kg twice a week for 10–12 weeks	Decreased expression of collagen type I and α-SMA, increased serum level of HGF, decreased serum level of TGF-β and NGF	[136]
Mouse ADSC transfected with miR-122, 1 × 10^6^ cells injected through tail vein	CCl_4_ (intraperitoneal), 1 mL/kg twice a week for four weeks	Decreased mature hepatic collagen type I α1, inhibition of cell proliferation and collagen maturation by HSC	[110]

ADSC: adipose-derived stem cell(s), ALT: alanine aminotransferase, AST: aspartate aminotransferase, CK: cytokeratin, MDA: malondialdehyde, PCNA: proliferating cell nuclear antigen.

**Table 6 cells-08-01339-t006:** Experimental liver fibrosis and pluripotent cells.

Cells/Dose/Route	Fibrosis Model	Main Results and Mechanisms of Action	Reference
Extracellular vesicles derived from human iPSC (iPSC-EVs) for in vitro experiments, animal models: 1.5 × 10^6^ murine iPSC-EVs by tail vein	Activated HSC in vitro, CCl_4_ and BDL models of fibrosis in mice	Downregulation of collagens, growth factors and TIMPs in HSCs, reduced proliferation and chemotaxis, reduced fibrosis in both models, downregulation of α-SMA, collagen-I α1, and TIMP-1	[168]
Hepatocyte-like cells derived from reprogramming of senescent mouse fibroblasts with Oct4, Sox2, Klf4, and the novel factor poly(ADPribose) polymerase 1 (Parp1), intra-splenic route	Non-alcoholic steatohepatitis (NASH) mouse model using a methionine/choline-deficient diet for four weeks	iPSC-derived hepatocyte-like cells were resistant to oxidative stress induced by hydrogen peroxide and lipid overload with fatty acids, increased IL-10 secretion, cell therapy attenuated macrovacuolar steatosis and restored liver function	[169]
Intra-splenic transplantation of 1 × 10^6^ hepatocyte-like cells derived from reprogrammed mouse embryonic fibroblasts	Acute and chronic liver injury was induced using different treatment protocols with CCl_4_ in BALB/c nude mice	Normalization of ALT and ALS levels in both models, reduced mRNA expression of fibrotic markers COL1α1, α-SMA, TGF-β1, and pro-inflammatory cytokines (TNF-α, IL-6, IL-1β), increased expression of antioxidant molecules NQO-1, HO-1, SOD-1, catalase and GST	[170]

BDL: bile duct ligation, GST: glutathione-S-transferase, HO-1: heme oxygenase 1, iPSC: induced pluripotent stem cell(s), NQO-1: NAD(P)H dehydrogenase (quinone 1), SOD-1: superoxide dismutase 1.

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
