# Peer review of "Mechanisms Underlying Cell Therapy in Liver Fibrosis: An Overview"

_cells, 2019, doi:10.3390/cells8111339_

Round 1

Reviewer 1 Report

This review covers a new aspect of therapy and might be of special interest of hepatologists.

In the first chapter, the authors have given only a short overview about cell types and mechanisms, which are involved in liver fibrosis. Since the mechanisms of liver fibrosis are comprehensively described in many other reviews, this short summary of background information is absolute fine. In the following chapters, the authors then discuss the function of cellular populations and the mechanisms in liver fibrosis under the aspect of cell therapy. Furthermore, the authors introduce the audience into the different cell populations and their specific application in cell therapy.

The review is well written, though minor editorial changes are recommended e.g. page 3 line 116 “used” should be corrected into “targeted” and in the table headings “cell therapy for liver fibrosis” should be changed into “cell therapy for inhibition of liver fibrosis “.  

Author Response

Response to Reviewer 1 Comments

The authors thank the reviewer for the kind evaluation and contributions to this work. The reviewer raised minor points which were easily corrected in the paper, as stated below:

Point 1: English language correction in Page 3 line 116: “used” should be corrected into “targeted”.

Response 1: Changes were done and can be seen in the line 137 of the revised version.

Point 2: In the table headings “cell therapy for liver fibrosis” should be changed into “cell therapy for inhibition of liver fibrosis”.

Response 2:  The authors found this mistake in the heading of table 2, which was corrected, and also in the title of Section 2 (line 134 of new version), which was changed to “Cell types used for inhibition of liver fibrosis”. Please let us know if all headings should be changed.

The authors appreciate all considerations and remain available for further questionings, so that the paper is disclosed under international scientific publishing standards and guidelines.

Reviewer 2 Report

The review by Pinheiro et al. (Cells 615229) has been prepared to offer an up-to date overview of actual state of cell therapy in chronic liver disease, with a focus on the anti-fibrotic effect of the different strategies that have been employed in this specific area of research.  The present review is, in  my view, an excellent and well balanced one and I personally do not see any major weakness in the manuscript prepared by Authors. I could offer just very few and minor points.

- paragraph starting at line 100:  please, cite here experimental dietary protocols for induction of progressive NAFLD/NASH, since this chronic liver disease is rapidly emerging in humans as the most relevant now and in the near future;

-  paragraph starting at line 485: there are at least two more studies suggesting a pro-fibrogenic role of MSC recruited from BM or transplanted  to be cited (see  Forbes SJ et al. Gastroenterology. 2004;126:955-6, di Bonzo LV et al., Gut. 2008;57:223-31);     

-  title of subsection at line 649: please, add to the title of this subsection       “……or hepatic progenitor cells”.

Author Response

Response to Reviewer 2 Comments

The authors thank the reviewer for the nice evaluation and contributions to this work. The reviewer raised minor points which were addressed in the revised paper, as stated below:

Point 1: Paragraph starting at line 100:  please, cite here experimental dietary protocols for induction of progressive NAFLD/NASH, since this chronic liver disease is rapidly emerging in humans as the most relevant now and in the near future.

Response 1: The authors have written further considerations on this important topic, adding a new paragraph and respective references (lines 100-120).

Point 2: Paragraph starting at line 485: there are at least two more studies suggesting a pro-fibrogenic role of MSC recruited from BM or transplanted  to be cited (see  Forbes SJ et al. Gastroenterology. 2004;126:955-6, di Bonzo LV et al., Gut. 2008;57:223-31).

Response 2: These are classic papers on this issue that were missed to be cited in the present work. The authors thank for this reminding and added these and a review paper to the references and text, in the paragraph starting at line 507 in the revised version.

Point 3: Title of subsection at line 649: please, add to the title of this subsection “……or hepatic progenitor cells”.

Response 3: Title corrected in the new version of the paper, now in the line 682.

The authors appreciate all important considerations regarding citations, and remain available for further questionings, so that the paper is disclosed under international scientific publishing standards and guidelines.
